# Panic Detection Using Machine Learning and Real-Time Biometric and Spatiotemporal Data

Ilias Lazarou * , Anastasios L. Kesidis, George Hloupis and Andreas Tsatsaris

Department of Surveying and Geoinformatics Engineering, University of West Attica, 12243 Athens, Greece
* Correspondence: ilazarou@uniwa.gr

**Abstract:** It is common sense that immediate response and action are among the most important terms when it comes to public safety, and emergency response systems (ERS) are technology components strictly tied to this purpose. While the use of ERSs is increasingly adopted across many aspects of everyday life, the combination of them with real-time biometric and location data appears to provide a different perspective. Panic is one of the most important emergency indicators. Until now, panic events of any cause tend to be treated in a local manner. Various attempts to detect such events have been proposed based on traditional methods such as visual surveillance technologies and community engagement systems. The aim of this paper is twofold. First, it presents an innovative multimodal dataset containing biometric and spatiotemporal data associated with the detection of panic state in subjects that perform various activities during a certain period. For this purpose, time-enabled location data are combined with biometrics coming from wearables and smartphones that are analyzed in real-time and produce data indicating possible panic events that are geospatially described. Second, the proposed dataset is used to train various machine learning models, and their applicability to correctly distinguish panic states from normal behavior is thoroughly examined. As a result, the Gaussian SVM classifier ranked first among seven classifiers, achieving an accuracy score of 94.5%. The dataset was also tested in a deep learning framework, achieving an accuracy level of 93.4%. A long short-term memory approach was also used, which reached a top accuracy of 94%. Moreover, the contribution of the various biometric and geospatial features is analyzed in-depth to determine their partial importance in the overall panic detection process. This is moving towards the creation of a smart geo-referenced ERS that could be used to inform the authorities regarding a potentially unpleasant event by detecting possible crowd panic patterns and helping to act accordingly, getting the information right from the source of the event, the human body. The proposed dataset is freely distributed to the scientific community under the third version of GNU General Public License (GPL v3) through the GitHub platform.

**Keywords:** panic detection; biometrics; machine learning; deep learning; classification; real-time data; emergency response systems; geospatial data

## 1. Introduction

Two of the most important pillars of social stability in every society are public health and public safety. Undoubtedly, the immediate operational response in public health or public safety emergency is more than crucial, and organizations of global reach, as well as nations themselves, integrate such mechanisms in their broader strategy [1,2]. It has been observed that during recent years the development of systems that target informing the authorities on-time about emergencies, has been exponentially increased [3].

Emergency response systems (ERS) are integrated solutions that handle urgent and severe events [4]. They have benefited from the evolution of information technology, which has resulted in increased responsiveness and effectiveness [5]. The wide range of online available sensors allows scientific decisions to be made regarding the emergencies based on real-time data.

Panic and its proper detection is an application field that undoubtedly would benefit from ERSs. Attempts to model and analyze panic behavior to detect, for example, crowd escape patterns, date back to 2000 when, for instance, Helbing et al. in [6] used a model of pedestrian behavior to investigate the mechanisms of (and preconditions for) panic and jamming by uncoordinated motion in crowds. In the recent literature, there are numerous studies as well as systems in production that deal with panic detection based on CCTV (Closed Circuit Television) technologies. They involve surveillance techniques that collect visual data in terms of still images and/or video sequences to analyze human behavior either of individuals or groups of people. For instance, Hao et al. [7] in their research study propose a brand-new approach to detect crowd panic behavior based on optical flow features. The very first step is a video capturing which is usually obtained from standard video sources such as online CCTV or offline video databases. Once features are extracted from the videos, the values are modeled and histograms or probability models are built to analyze the crowd or individual behaviors. In another view, Ammar et al. [8] describe an online and continuous surveillance system of a particular public place using a fixed camera on the one hand, and a methodology for real-time analysis of the captured images on the other hand. The system is based on a deep learning framework where a long short-term memory (LSTM) approach is adopted that is capable of learning order dependence in data sequences. LSTM has proved to be considerably successful in remembering correlations among temporal events and in predicting a future value given historical data [9].

A special category of such systems is based on the user's intervention in the reporting of an emergency event. A drawback of such approaches, which are broadly referred to as community engagement methods, is that during an emergency it is highly unlikely that people will give priority to reporting the event, instead of running away. Sufri et al. [10] state that ERSs are a key part of disaster preparedness [11]. They are considered an essential component in providing timely and effective information to individuals and communities so that immediate, appropriate and effective responses can be taken to reduce potential injury and death, and damage to property and livelihoods [12–14].

In a different type of approach, other research studies use wearable devices to collect biometric data and analyze them for stress detection. Wearables provide a mature way of collecting detailed and real-time data as there is access to numerous technologically advanced sensors giving valuable information such as heart rates, heart rate variability, oxygen levels, and many more depending on the kind of the device. Furthermore, by pairing them with the latest communication methods, such as 5G smartphone sensor capabilities, one could say that there is a combination of real-time data that can be a game changer in this research field. Liu et al. [15], after analyzing the literature on crowd evacuation based on data extracted from the Web of Science by applying bibliometric approaches, conclude that research on systems, quantitative analysis, and visualization studies on crowd evacuation is still a developing field [16–20]. Chan-Hen Tsai et al. [21] use data from wearables to predict how possible is for a person to develop panic attack disorder in the future based on time-series data. Specifically, this study aims to provide a 7-day PA (Panic Attack) prediction model and determine the relationship between a future PA and various features, including physiological factors, anxiety and depressive factors, and the air quality index (AQI). In another study, Rubin et al. [22] intended to predict oncoming panic attacks and to deliver in-the-moment interventions on a smartphone device. These kinds of intervention are intended to reduce symptom severity by enabling a user to respond to approaching panic episodes. Next, Kutsarova and Matskin propose a system that combines mobile crowdsensing and wearable devices to manage alarming situations [23]. This work is based on an existing crowdsensing system named CrowdS. A smartwatch application that utilizes the device's sensors to detect abnormal events was created. Then there were two approaches for integrating the smartwatch with the CrowdS platform. A direct connection through the Internet, and a connection through a smartphone by pairing a smartwatch application via Bluetooth.

Lastly but most interestingly, Alsalat et al. [24] attempt to detect human panic based on wearable devices using machine learning techniques to distinguish between the stressed and calm state of an object. Their team has constructed a method that collects data from the user, classifies it into stress or calm and, if stressed, passes the data to a server for post-processing and mapping. They have experimented with machine learning and deep learning approaches and reported high-accuracy results. Unfortunately, their dataset is not publicly available.

The aim of this paper is twofold: first, it proposes a new type of geospatial (point geometry) dataset consisting of biometric and spatiotemporal data for stress detection, and second, it investigates the use of machine learning techniques to develop a prediction model for stressful events. This is achieved by collecting non-personalized data from wearable devices and smartphones and using them to train a properly selected machine learning model. The new dataset consists of biometric features such as heart rate and heart rate variability; spatiotemporal and motion data such as location coordinates (longitude, latitude), activity, speed, and steps; and also descriptive data such as gender, age, and weight. A derivative biometric characteristic is also proposed that quantifies the deviation of a current heart rate measurement against the heart rate moving average of various time windows. Ranking tests highlighted the importance of the proposed derived feature, while the experimental results show that it significantly improves the classification accuracy of the various machine and deep learning models. Visualization experiments further demonstrate that this approach can be the basis of an efficient panic detection system.

The proposed approach tackles the weaknesses that arise in other alternative methods regarding panic detection and provides an efficient framework to overcome them. Specifically, panic events detected by visual surveillance technologies are spatially limited by the range of the visual equipment. Similarly, community engagement systems rely heavily on permanent human awareness and immediate personal response. On the other hand, the real-time biometric and spatiotemporal nature of the data in the proposed approach is spatially unrestricted and information is flawlessly transmitted right from the source of the event, the human body. Although the biometric measurements are highly affected by changes in the emotional and psychological status of a subject, in the proposed approach the scope is to detect the subject's panic state without distinguishing between the various sources of stress (medical, psychological, emotional, etc.). The proposed multimodal dataset, which combines a bundle of geospatial, biometric, and motion perspectives, provides scientists with the ability to apply enriched methodologies and scale the analysis starting from an individual level and potentially extend it to crowd level. Having the machine learning classifiers trained on such a rich dataset for stress detection, depending initially on the biometric and motion aspect of the features, enables the further examination of the phenomenon. This is to compare the spatiotemporal correlation of the subjects targeting the detection of possible unforeseen events at a crowd level.

## 2. Panic Behavior and Sensing

### 2.1. Panic

Panic is a phenomenon generally studied in psychology and human sciences and is often identified by its consequences. It is a sudden, strong feeling of anxiety or fear that prevents reasonable thought and action and may spread to influence many people [25]. It can be felt by an individual or simultaneously in a group or a crowd and is characterized by the regression of mentalities to an archaic and gregarious level, leading to primitive reactions of hopeless jumps, indiscriminate agitation of violence or collective suicide, as described in [26].

Panic can be studied by two different aspects, namely, its physical features and its psychological characteristics [27]. Regarding the physical features, Helbing et al. [6] as well as Bracha [27] refer to the fight and flight response. Helbing states that "the characteristic features of escape panics can be summarized as follows: (1) People move or try to move considerably faster than normal. (2) Individuals start pushing, and interactions among

people become physical in nature. (3) Moving and, in particular, the passing of a bottleneck becomes uncoordinated. (4) At exits, arching and clogging are observed. (5) Jams build up. (6) The physical interactions in the jammed crowd add up and cause dangerous pressures which can bend steel barriers or push down brick walls. (7) Escape is further slowed by fallen or injured people acting as 'obstacles'. (8) People show a tendency towards mass behavior, that is, to do what other people do. (9) Alternative exits are often overlooked or not efficiently used in escape situations".

On the other hand, the psychological characteristics are tied to specific bio-signals as stated by Haag et al. [28]. When we are frightened our heart races, breathing becomes rapid and mouth becomes dry, muscles are tense, our palms become sweaty, and we may want to run. These bodily changes are mediated by the autonomic nervous system, which controls the heart muscle, smooth muscle, and exocrine glands [29]. The autonomic nervous system itself can be divided into sympathetic and parasympathetic divisions. Both operate in conjunction with each other and with the somatic motor system to regulate most types of behavior, whether in normal or emergencies. Although several visceral functions are controlled predominantly by one or the other division, and although both the sympathetic and parasympathetic divisions often exert opposing effects on innervated target tissues, it is the balance of activity between the two that helps maintain an internal stable environment in the face of changing external conditions. Certain emotions can affect this balance and can result in a wide variety of bodily reactions comparable to the ones described above [30].

*2.2. Mobile Crowdsensing*

Mobile crowdsensing, first introduced by Ganti et al. in [31], is a technique that uses smartphones, tablets, computers, and wearable devices to collect required data and extract information to measure, analyze, estimate or infer any processes of common interest [32]. It takes advantage of IoT technology, a market growing at a rapid pace. People using these devices play the role of real-time sensors that feed the mechanisms with their own real-time data.

Currently, the actual crowd involvement and sensing deals with location referencing and biometric data monitoring. Saquib et al. [33] note that heart rate, or pulse, is one of the vital signs used to measure the basic functions of the human body. Other biometric data include electro-dermal activity, electromyogram, respiration, electrocardiogram, blood volume pulse, heart rate variability, skin temperature, and pupil diameter, as Nath et al. state in [34]. These biometric data are collected from an IoT device, such as a smartwatch, by applying a method called photoplethysmography [35]. It is a non-invasive low-cost technique that makes measurements at the skin surface detecting blood volume changes in the microvascular bed of the tissue.

In accordance with such biometric data, spatiotemporal information regarding the physical location of the subject including latitude, longitude, altitude, speed, steps, and current timestamp, are also collected. A collection of features such as the above-mentioned form the basis of the proposed machine learning approach described in the next sections.

## 3. Methodology

The scope of the proposed methodology is to transform the gathered measurements (biometric and spatiotemporal data) into valuable information in order to expose hidden patterns that possibly reveal panic behavior. Starting from the user's endpoint, the workflow begins from the wrist where an application running on the wearable device monitors the real-time biometric footprint regarding data such as heart rate and heart rate variability. At the same time, a paired application running on an Android smartphone collects GPS location coordinates (longitude, latitude), time data, user activity, speed and steps. Following a time interval of one second, all this information is bundled together into a single UDP packet and is sent encrypted to a server through GSM network. On the server side, a Java code receives the UDP packets, decrypts the information and constructs point geometries having all the above-referenced characteristics as attributes. This procedure enables the

collection of real-world biometric and spatiotemporal data that are the basis for creation of the proposed dataset. A large collection of such measurements is used to train the machine learning classifier to properly distinguish panic states from normal behavior, as shown in Figure 1. The initial real-world data were collected by two individuals, one male and one female, using the applications and aforementioned hardware configuration. The data collection process took place in a neighborhood in Paleo Psychiko, Athens, Greece. As it would be difficult to produce real and sudden stress conditions, the subjects coordinated their movements and reactions in order to simulate them.

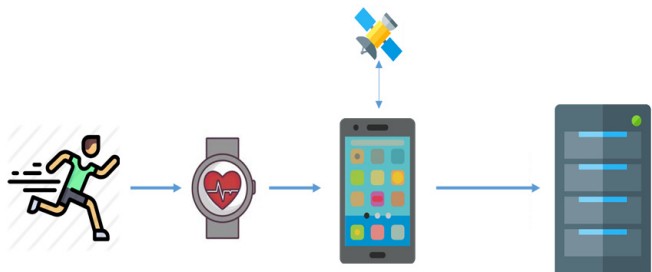

**Figure 1.** System workflow.

Figure 2 demonstrates the real-world data collection of the male participant. The subject (person) is on foot until suddenly something happens where he enters a panic state and starts running. The subject keeps running even when the panic state is over and, finally, he gradually slows down and walks till the end of his path. Figure 3 depicts the corresponding biometric and spatiotemporal measurements collected during the experiment. It can be seen that the heart rate inclines aggressively during running, while the heart rate variability follows an opposite path and declines significantly. All the measurements reflect the state changes and also are affected as the subject attempts to escape in panic. Once the subject leaves the stressed state, the biometric and spatiotemporal measurements start gradually to decline and return to normal levels.

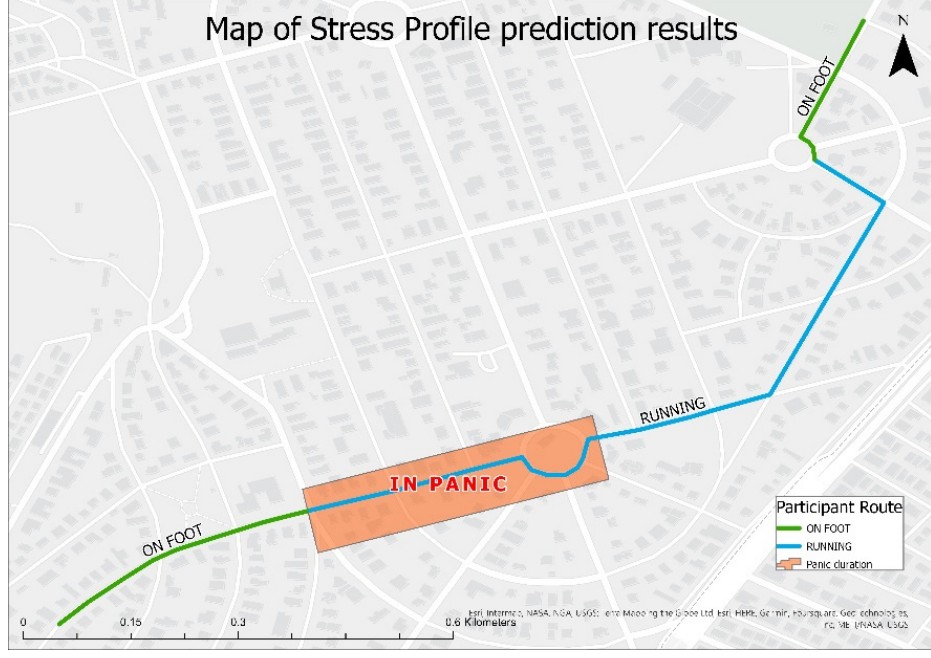

**Figure 2.** Map representation of the route where the male participant collected real-world data.

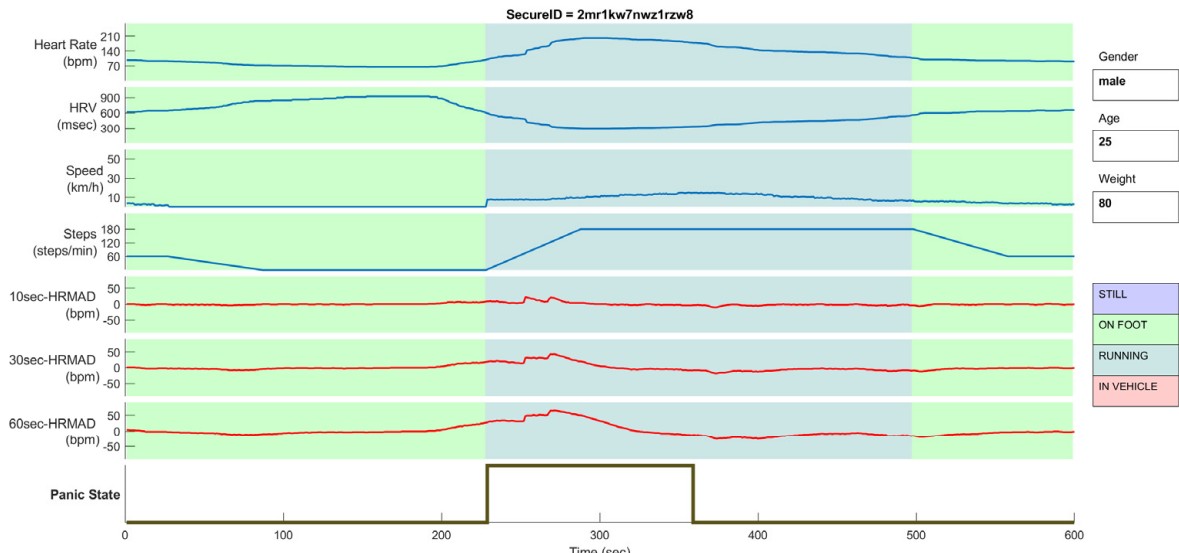

**Figure 3.** Biometric and spatiotemporal data collected during the experiment by the male participant.

### 3.1. Dataset

In this section, a dataset is proposed that consists of 27 different subjects that are monitored during a short time frame. Two of the 27 subjects are actual humans that used the wearable and the smartphone and captured real-world data using the accompanying applications. The data regarding the rest of the subjects were artificially produced. Their biometric and geospatial data are depicted per second for 10 min resulting in a set of 600 measurements per subject. In most cases, a panic event is simulated that affects these measurements. However, in three of them, there is no panic event. This is on purpose to capture the variability of the observed data in both calm and stressed states. The collected data can be summarized as follows:

1. Biometric (from wearable): heart rate, heart rate variability
2. Spatiotemporal (from smartphone): location coordinates, activity, speed, steps
3. Descriptive (from wearable): gender, age, weight
4. Secure ID (from smartphone)

To achieve a realistic set of measurements, the values of the several features are carefully determined based on studies that provide such relevant information. For instance, according to Forbes Health [36], normal heart rate varies according to age. Individuals between one and four years old have normal resting heart range between 80 and 120 to 130 bpm. Individuals in the age group five to nine years old range from 70 to 100 or 115 bpm. Normal resting heart rate in ages 10 and above reaches 60 to 100 bpm. Athletes belong to a separate category as their range is 40 to 60 bpm. Moreover, American Centers for Disease Control [37] states that the maximum heart rate is about 220 bpm (beats per minute) minus the subject's age. Additionally, the target heart rate during activities of moderate intensity is about 50–70% of the maximum heart rate, while during vigorous physical activity it is about 70–85% of the maximum. For example, the maximum heart rate for the age group 20 to 40 ranges between 180 and 200 bpm. The age group 45 to 60 has its values around 160 to 175, and ages 65 years and above have 150 to 155 bpm.

For the collection of the raw biometric and positional data, a Samsung Galaxy Watch wearable, as well as a Samsung Galaxy A70 smartphone were used. The biometric and spatiotemporal features of the dataset are described in detail as follows:

Heart Rate: It is the main biometric feature and is measured in beats per minute (bpm).

Heart Rate Variability (HRV): This is the time difference between two consequent heart beats and is measured in milliseconds (ms). The more intense the heart beats, the smaller the time difference between the beats.

Location Coordinates: Latitude and longitude of the location.

Activity: This information comes from the gyroscope and accelerometer sensors of the smartphone and defines whether someone is standing still, walking, running, or is in a vehicle.

Speed: It is provided by the smartphone's GPS and is measured in km/h.

Steps: This metric is also provided by the smartphone and is measured in steps per minute.

Gender, Age, Weight: This demographic information is useful since it affects biometric measurements. For instance, a 78-year-old woman weighing 60 kg has a different biometric footprint while in panic than a 20-year-old man weighing 85 kg. In total, the proposed dataset includes 14 male and 13 female subjects with varying age and weight values.

In addition to the above raw data, a derived feature named heart rate moving average deviation (HRMAD) is introduced. It encloses a temporal effect on the dataset that is based on a time window regarding heart rate values of the past. It indicates that someone has suddenly developed high measurements of the heart rate which could imply sudden panic conditions. Typically, the mean value of the last minute's heart rate should be around 5–10 bpm based on the assumption that it slightly varies from the resting heart rate levels. Considering that a subject that walks in calm condition logs heart rate values of 70 or 80 bpm, he/she would have a heart rate mean between 70 and 80 and each consequent measurement in a calm state would eventually be in that range as well. On the other hand, a sudden event that causes panic would exaggerate the heart rate possibly beyond 150 bpm which means that the difference from the previous measurements would be very high. The proposed feature is calculated in three different time windows of 10, 30, and 60 s (HRMAD10, HRMAD30, and HRMAD60), respectively. It provides an indication of how much the current heart rate measurement deviates from a moving average of a specific time window in the past. The time window acts as a smoothing technique where potential residuals and deviations are absorbed by the averaging process. In this manner, a possible, e.g., high-valued, current measurement may not deviate significantly from a short time window of similarly high-valued measurements; however, it will heavily deviate from a long time window where potential "normal" (non-high) measurements are also included in the calculation of the moving average.

It should be noticed that the dataset includes a timestamp value that provides a detailed chronological description of each separate measurement collected per second. Table 1 summarizes the description of the features in the dataset.

**Table 1.** Dataset features description.

| Feature | Type | Values | Units |
| --- | --- | --- | --- |
| Heart Rate | Numeric | 56 to 246 | bpm |
| Heart Rate Variability | Numeric | 244 to 1071 | msec |
| Activity | Categorical | Still, on foot, running, in vehicle | - |
| Speed | Numeric | 0 to 58.3 | km/h |
| Steps | Numeric | 0 to 240 | steps/min |
| Gender | Categorical | male, female | - |
| Age | Numeric | 20 to 84 | years |
| Weight | Numeric | 50 to 90 | kg |
| HRMAD10 | Numeric | −24 to 40 | bpm |
| HRMAD30 | Numeric | −42 to 58 | bpm |
| HRMAD60 | Numeric | −55 to 86 | bpm |

### 3.2. Dataset Scenarios

The proposed dataset attempts to depict various scenarios that simulate the behavior of subjects under stress and/or panic conditions without differentiating between the reasons that caused them, which can have either biological or psychological basis, or come from a sudden unforeseen event that happened nearby stressing the subject. Table 2 provides

details about six different scenarios that are utilized in the dataset as well as three more scenarios that correspond to calm-only (no panic) conditions.

**Table 2.** Dataset scenarios.

| Scenario | Description | Example |
|---|---|---|
| **Panic Scenario 1** | While being still, the subject starts running suddenly in panic as if an escape attempt takes place. | The subject rests when suddenly an alarming situation occurs. It stands up immediately and starts running out in the street. |
| **Panic Scenario 2** | While being on foot, the subject starts running suddenly in panic as if an escape attempt takes place. | The subject walks in the street and suddenly starts running because an attack takes place nearby. |
| **Panic Scenario 3** | While running for exercise the subject starts running in panic and its biometric characteristics start to deviate. | The subject runs at night for exercise when suddenly a wild animal starts chasing him. The subject begins to accelerate in an attempt to escape. |
| **Panic Scenario 4** | While being still, the subject's biometric characteristics start to deviate but the subject freezes to its location. | The subject is still, and an unpleasant event happens. It panics and freezes in its location. When it calms down, it walks away from the scene. |
| **Panic Scenario 5** | While being on foot, the subject's biometric characteristics start to deviate but it keeps walking as if it is not able to run. | The subject is walking when an attack takes place nearby. It panics but it remains in a walking state because it is slightly wounded. |
| **Panic Scenario 6** | While being in a vehicle, the subject's biometric characteristics start to deviate, it stops the vehicle and starts running. | The subject is driving when an explosion happens nearby. All the vehicles are immobilized so it leaves the vehicle and escapes the scene running. |
| **Calm Scenario 1** | While being still, the subject suddenly starts running. | The subject is resting but suddenly remembers to do something, so it starts running. |
| **Calm Scenario 2** | While being on foot, the subject's biometric characteristics start to deviate but it keeps walking. | The subject is walking on the street but suddenly develops palpitation due to cardiac causes. |
| **Calm Scenario 3** | While driving, the subject stops the vehicle and starts running. | While the subject is driving a car, it parks it and starts running to catch the bus. |

Each of the above-mentioned panic scenarios is provided in four different cases depending on the subject's descriptive characteristics. These are young/old as well as male/female. Thus, the dataset includes 27 cases, that is, 24 panic cases and three calm ones. Figure 4 depicts a panic case that belongs to Scenario 1. For the first 100 s, the subject appears to be calm in a still state. The motion indexes are flat to zero and the heart indexes fall inside the resting ranges. Then suddenly an alarming situation occurs, and the subject starts to walk for around 20 s preparing for its escape attempt. Next, it speeds up and starts running for the next 22 s. After that, it hides in a safe place and waits for 55 s. Once it is safe it leaves its refuge and continues running away for the next 40 s. All this activity marks its stress period which falls between the 100th and the 250th second, that is, 2 min and 30 s. During this period, the biometrics such as heart rate, heart rate variability, and HRMAD of 10, 30, and 60 s start to deviate gradually, while the spatiotemporal data such as speed, steps, and state are also affected. Specifically, the heart rate peaks at 152, the HRV falls to 395 and the HRMAD10, HRMAD30, and HRMAD60 peak at 20, 35, and 55 bpm, respectively. As far as the motion characteristics are concerned, the speed reaches 9.9 km/h and the steps climb to 87 steps per minute. After the 254th second the subject exits the stressed state and walks normally for about 200 more seconds when it finally stops. The biometric and motion values are returning back to normal levels.

In Figure 5 a scenario different than the previous one is demonstrated that involves driving activity. The subject begins its route by driving a vehicle at a speed of around 40 km/h. The biometric values are within a normal range for this activity, as the heart rate is about 90 bpms, HRV at 650 ms, and HRMAD indexes at a range of −2 to 2, respectively. The motion data are in a calm state while steps per minute are flat to zero as the subject is driving. All these till the 50th second when something happens on the road and everyone panics. The subject, still in its vehicle, attempts to escape the scene by driving for the next 2 min. At this point, it realizes that it cannot accelerate more, as one can see in the diagram where speed remains constant. The subject decides to stop the vehicle and continue on foot.

At the 190th second, the activity switches to still as the vehicle stops and remains still for the next few seconds. Then the subject leaves the place on foot for the next 13 s and starts running to escape for the following 20 s. Till this point, the measured data show that the subject is in a stressed condition that lasted for 3 min since the panic event occurred. During this period the heart rate peaked at 152 bpm, HRV fell to 395 and HRMAD climbed to 12, 26, and 36 bpm, respectively. Regarding the spatiotemporal data, speed while running reached 6.6 km/h, and steps hit a high of 58 per minute. After the end of the panic period, the subject continued to run for some seconds but in a calm state and then walked for about 2.5 minutes before stopping completely.

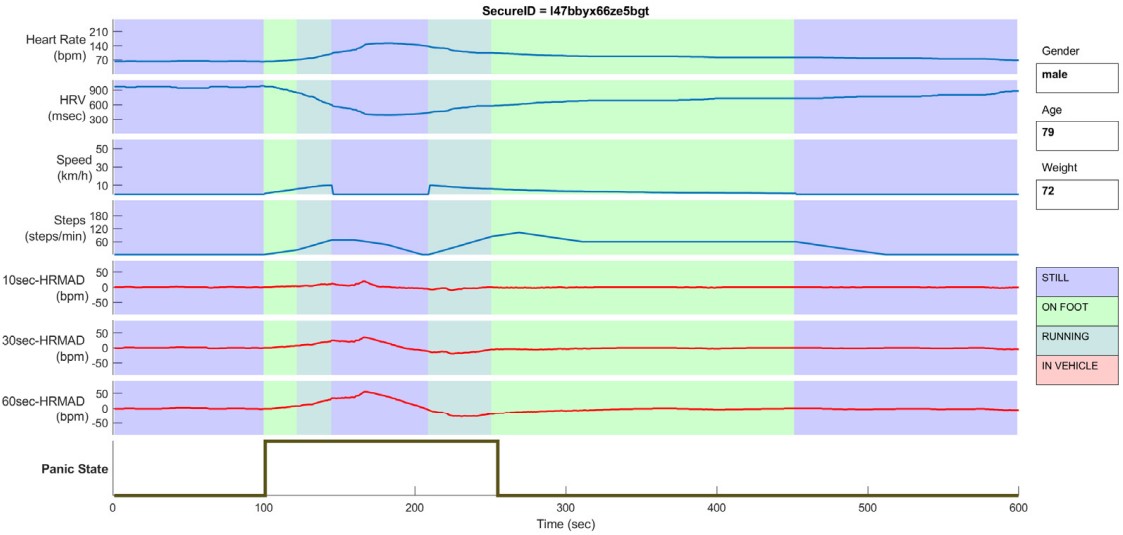

**Figure 4.** Scenario 1 example.

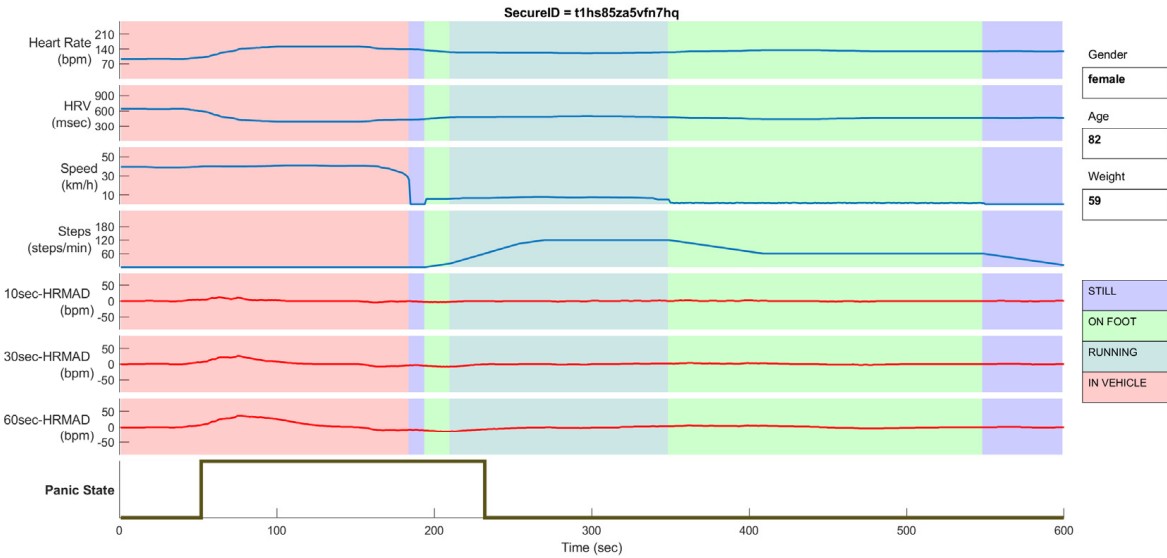

**Figure 5.** Scenario 6 example.

In all the previous examples, each measurement represents a subject either in a calm or stressed state. In the next section, machine learning models are involved to classify the measurements into these two classes. While the model will be trained offline using the dataset described above, the ultimate purpose of this approach is to efficiently detect the panic state among biometric and spatiotemporal measurements on a real-time basis.

## 4. Experimental Setup and Results

### 4.1. Machine Learning Classifiers

This section describes how the proposed dataset is used to train various machine learning models and examines their applicability to correctly distinguish panic states from normal behavior. Moreover, the contribution of the various biometric and geospatial features is analyzed in-depth to determine their partial importance in the overall panic detection process. Multiple modern machine learning models are used for the classification process, namely, decision trees [38], logistic regression [39], Gaussian and kernel naïve Bayes [40], Gaussian SVM and SVM kernel [41], and boosted trees [42]. The cross-entropy is used as the cost function for the classification tasks. Table 3 summarizes the classification setup parameters for each model. In all experiments, the available dataset is split into 90% for training and 10% for testing purposes. All experiments are implemented in Matlab® and carried out on an Intel Core i7 @2.6 GHz with 16 GB of RAM.

**Table 3.** Classification setup parameters.

| Classifier Model | Hyperparameters |
| --- | --- |
| Decision Tree | Max number of splits: 100, Split criterion: Gini's diversity index, Surrogate decision splits: Off, Max surrogates per node: 10 |
| Logistic Regression | No hyperparameter options |
| Gaussian Naïve Bayes | Numeric predictors distribution: Gaussian, Categorical predictors distribution: MVMN |
| Kernel Naïve Bayes | Numeric predictors distribution: Kernel, Categorical predictors distribution: MVMN, Kernel Type: Gaussian, Support: Unbounded |
| Gaussian SVM | Kernel Function: Gaussian, Kernel scale: 0.61, Box constraint level: 1, Multiclass Method: One-vs-One, Standardize data: True |
| SVM Kernel | Learner: SVM, Number of expansion dimensions: Auto, Lambda: Auto, Kernel scale: Auto, Multiclass Method: One-vs-One, Iteration limit: 1000 |
| Boosted Trees | Ensemble method: Adaboost, Learner type: Decision Tree, Max number of splits: 20, Number of learners: 30, Learning rate: 0.1, Number of predictors to sample: All |

The effectiveness of the classifiers on the classification task is initially tested on the raw biometric and spatiotemporal data, namely, heart rate, heart rate variability, activity, speed, and steps. Table 4 shows the accuracy of the various models. It can be seen that the Gaussian SVM classifier achieves the highest accuracy score followed by the decision tree. On the contrary, Gaussian naive Bayes provides the poorest results.

**Table 4.** Classification results using raw features. Best result is shown in bold.

| Classification Model | Accuracy (Raw Features) |
| --- | --- |
| Decision Tree | 91.4% |
| Logistic Regression | 89.6% |
| Gaussian Naïve Bayes | 79.6% |
| Kernel Naïve Bayes | 80.3% |
| Gaussian SVM | **93.2%** |
| SVM Kernel | 90.6% |
| Boosted Trees | 90.9% |

The classification results are further improved when the proposed HRMAD feature is involved. Table 5 shows the accuracy for all three cases, namely HRMAD10, HRMAD30, and HRMAD60, respectively. It can be seen that almost all classifiers (except logistic

regression) improve their accuracy. The best result is once again provided by the Gaussian SVM. The most significant improvement occurs for kernel naïve Bayes, which raises the accuracy from 80.3% to 85.3%.

**Table 5.** Classification results using raw features and HRMAD. Best results are shown in bold.

| Classification Model | Accuracy (Raw Features + HRMAD10) | Accuracy (Raw Features + HRMAD30) | Accuracy (Raw Features + HRMAD60) |
|---|---|---|---|
| Decision Tree | 93.3% | 92.8% | 92.8% |
| Logistic Regression | 89.5% | 89.0% | 89.5% |
| Gaussian Naïve Bayes | 80.4% | 80.6% | 81.3% |
| Kernel Naïve Bayes | 83.7% | 84.3% | 85.3% |
| Gaussian SVM | **94.2%** | **94.4%** | **94.5%** |
| SVM Kernel | 91.8% | 92.3% | 94.1% |
| Boosted Trees | 92.3% | 93.3% | 93.9% |

We also investigated the classification performance from a feature selection perspective. Indeed, the question that arises is what level of accuracy can be achieved when features with low predictive power are removed. This is an important issue, when data collection is expensive or difficult, so a model that performs satisfactorily without some predictors may be preferred. For this purpose, three feature ranking tests were involved to estimate the significance of each predictor, namely, the minimum redundancy maximum relevance (MRMR), the chi-squared, and the ANOVA [43]. The MRMR algorithm finds an optimal set of features that is mutually and maximally dissimilar and can represent the response variable effectively. Similarly, the chi-square ($\chi 2$) distribution is a one-parameter family of curves commonly used in hypothesis testing, particularly the chi-square test for goodness of fit. Finally, analysis of variance (ANOVA) is a statistical formula used to compare variances across the means (or average) of different groups.

Table 6 summarizes the ranking results for feature selection. HRV and HRMAD60 achieve the top two ranking positions. Based on these ranking results, Table 7 shows the classification results when only HRV or HRV + HRMAD60 features are used. It can be seen that even with a single or a pair of top-ranked features the classifiers still provide reasonable results.

**Table 6.** Feature ranking position according to three ranking tests.

| Feature | MRMR | $\chi 2$ | ANOVA |
|---|---|---|---|
| HRV | 1 | 2 | 1 |
| HRMAD60 | 2 | 1 | 2 |
| Heart Rate | 4 | 3 | 3 |
| Activity | 3 | 5 | 6 |
| Speed | 6 | 4 | 5 |
| Steps | 5 | 6 | 4 |

**Table 7.** Classification results after feature selection. Best result is shown in bold.

| Classifier Model | Accuracy (HRV) | Accuracy (HRV + HRMAD60) |
|---|---|---|
| Decision Tree | 87.0% | **91.5%** |
| Logistic Regression | 86.1% | 88.2% |
| Gaussian Naïve Bayes | 86.6% | 87.3% |
| Kernel Naïve Bayes | 86.3% | 89.1% |
| Gaussian SVM | **87.0%** | 90.7% |
| SVM Kernel | 78.0% | 79.7% |
| Boosted Trees | 87.3% | 90.7% |

### 4.2. Deep Learning

In addition to the machine learning classifiers, we used the proposed dataset to train a deep neural network (DNN) as well as an LSTM recurrent neural network. The DNN architecture consists of a fully connected layer with an output size of 50, followed by a batch normalization and a RELU thresholding layer. At the end of the network, a fully connected layer classifies the data into the two classes by applying a softmax function. The Adam optimizer is used with an initial learning rate of 0.001 while the mini-batch size is set to 16. In this approach, four cases were tested. The first one includes only the raw features (heart rate, HRV, speed, steps, and activity), while the other three cases include one of the three HRMAD time windows (HRMAD10, HRMAD30, HRMAD60). Table 8 presents the corresponding classification results.

**Table 8.** DNN classification results. Best result is shown in bold.

| Approach | Accuracy |
|---|---|
| Raw Features Only | 91.5% |
| Raw Features + HRMAD10 | 91.8% |
| Raw Features + HRMAD30 | 92.3% |
| **Raw Features + HRMAD60** | **93.4%** |

The LSTM model is a type of recurrent neural network that can learn an internal representation of time series data. In order to be comparable with the previous experiments that involve the proposed HRMAD feature, again four cases were examined that differ by the sequence length applied. Specifically, in the first case only the current time sample is used while in the other three cases the time steps length is 10, 30 and 60 samples, respectively. The LSTM network architecture consists of a sequence input layer of size five, which corresponds to the number of raw features. It is followed by an LSTM layer with 100 hidden units that leads to a fully connected layer with two classes where the softmax function is used. The Adam solver is used, with a learning rate of 0.001, a gradient threshold of one and a maximum number of 50 epochs. Table 9 summarizes the classification results for the LSTM network.

**Table 9.** LSTM results. Best result is shown in bold.

| Sequence Length | Accuracy |
|---|---|
| 1 sample | 90.6% |
| 10 samples | 90.8% |
| 30 samples | 91.4% |
| **60 samples** | **94.0%** |

Considering the results in both deep learning approaches, it can be seen that the HRMAD feature manages to noticeably improve the accuracy when compared to the case where only the raw features are applied. Moreover, in both architecture, increasing the size of the HRMAD time window (which in the case of LSTM is expressed as the sequence length) results in increased accuracy. Comparing the two approaches, the LSTM provides the best overall accuracy of 94.0%. However, this is lower than the classification accuracy of 94.5% achieved by the Gaussian SVM.

### 4.3. Discussion

The motivation of the present work is to utilize a machine learning approach to successfully predict the panic state of a subject based on a set of biometric and spatiotemporal features that are coming from wearables and smartphones, with the ultimate goal of providing the basis for analyzing the spatiotemporal correlation of the data at a crowd level in a subsequent stage. To demonstrate whether the proposed framework can remit the issue or not, the Gaussian SVM classifier is applied using the five raw features combined with the

derived HRMAD60 data. This setup has shown the highest accuracy among all classifiers as shown in Table 4. In the first example, the subject (female, aged 77, 60 kg weight) iterates through several states starting from a still position, then walking, running, walking again, and finally standing still again, as shown in Figure 6. Her biometric and positional data vary significantly during these state transitions. For instance, Her heart rate ranges from 70 to 186, her HRV ranges from 323 to 909, speed is up to 9.6 and her steps are approaching 120 steps per minute. The calculated HRMAD60 values are in the range of 55 to −33. In Figure 6 all the biometric and spatiotemporal data are depicted for a specific moment in the 200th second. Even though the feature values vary significantly, the classifier accurately detects the panic state showing only a negligible error at the end of the stress period.

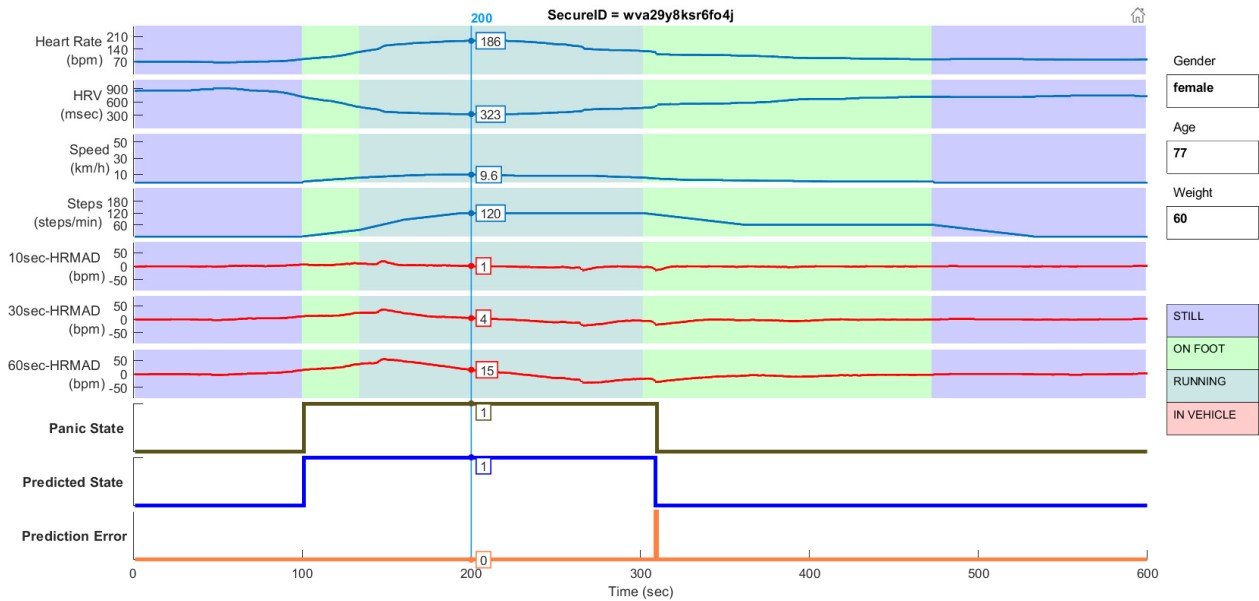

**Figure 6.** Successful panic prediction example.

On the contrary, an erroneous case is shown in the example of Figure 7. It depicts a case where a female subject, aged 20, weighing 50 kg is in a vehicle when an unpleasant event causes panic nearby. She keeps driving for a while and then stops the car and starts running. Her heart rate values are between 96 and 168, her HRV is between 357 and 625, and speed varies significantly from 58.3 down to 13.6 while the steps range from zero up to 180. The calculated HRMAD60 values are in the range of 45 to −18. The several biometric and spatiotemporal data can be seen in Figure 7 for the 230th second. The classifier correctly detects the beginning of the panic state; however, it outputs an erroneous calm state from seconds 243 to 283 as well as for a shorter period between seconds 307 to 312. In this case, compared to the previous example, the classifier has failed to correctly detect the whole duration of the panic period but, still, it has managed to identify it in its largest percentage. This behavior means that, in a completed system, such a case would flag three consecutive individual panic events instead of one. Thus, despite that the object is actually in a constant panic state for about 3 min, the classifier erroneous indicates that the person rapidly (and even abnormally) switches from calm to panic states three consecutive times in the same period.

Lastly, in Figure 8 one can see a map result of the classifier on a new set of data where the points have been tagged either as stressed or calm and depicted on the map using green and red colors, respectively.

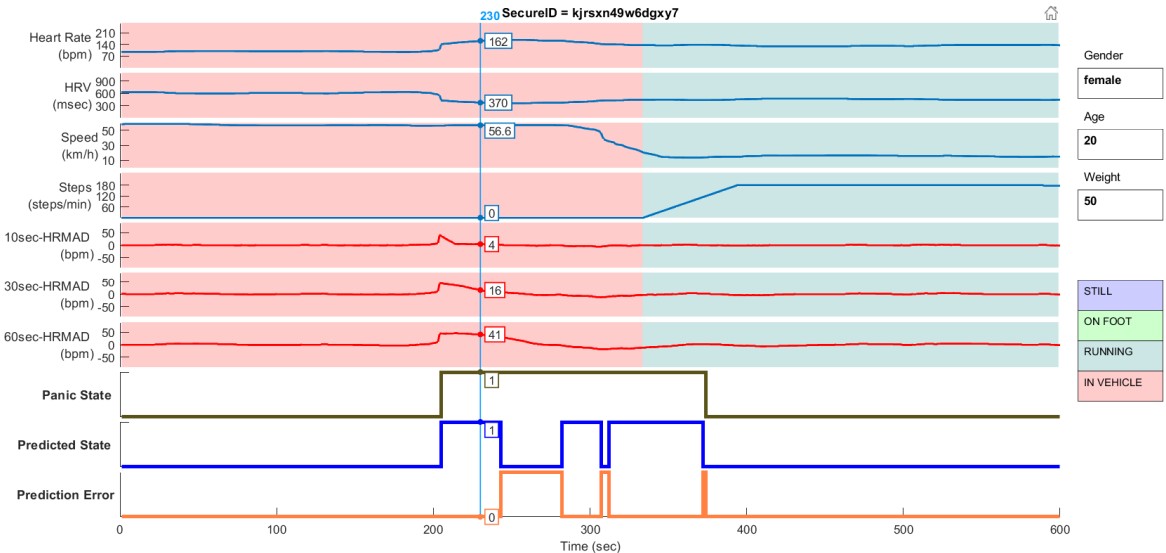

**Figure 7.** Partially erroneous panic prediction example.

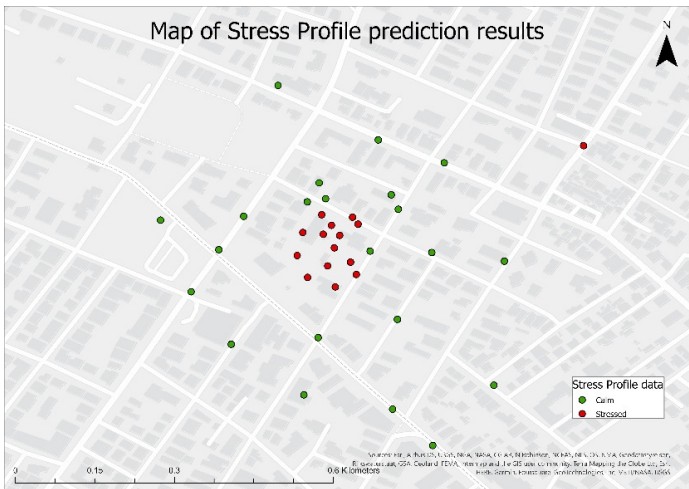

**Figure 8.** Map of stress profile prediction results from the selected classifier.

These visualization results demonstrate two points. First, a careful selection of raw biometric and spatiotemporal data in addition to calculated features such as the proposed HRMAD60 can provide high-level discrimination results when applied to a proper machine learning model such as Gaussian SVM. Second, a well-trained classifier can be the basis for a real-time panic prediction system. Furthermore, on a larger scale, having a crowd streaming this kind of real-time data leads to a crowdsourcing type of information system being built. This is moving towards the creation of a smart geo-referenced ERS that could be used to inform the authorities regarding a potentially unpleasant event by detecting possible crowd panic patterns and helping to act accordingly.

## 5. Conclusions

While the use of the ERSs is increasingly adopted across many aspects of everyday life, the combination of them with real-time biometric and location data appears to provide a different perspective. This work is an attempt to introduce a new way to detect abnormal behavior and panic by using these data. Its nature is twofold. First, a multimodal geospatial dataset has been created combining multiple features that refer to several activities in the field. The proposed dataset contains different scenarios, each of which involves a panic state of the subject that occurs for some seconds. The proposed dataset (will be) freely

distributed to the scientific community under the third version of GNU General Public License (GPL v3) through the GitHub platform. Second, this dataset acted as input to machine learning classifiers to successfully predict the subject's panic state based on raw biometric and spatiotemporal data. A derivative biometric characteristic is also proposed that quantifies the deviation of a current heart rate measurement against the heart rate moving average of the last 10, 30, and 60 s. Ranking tests highlighted the importance of HRMAD among the other raw features. In accordance, the experimental results showed that HRMAD significantly improved the classification accuracy of the various machine-learning models, as well as the LSTM recurrent neural network of the deep learning field. Regarding machine learning, the best classification results are reported by a Gaussian SVM classifier, reaching an accuracy of 94.5%. On the LSTM side, the network achieved an initial accuracy of 91.52% using the raw features, which was later improved to 93.42% with the inclusion of the proposed HRMAD feature. Visualization experiments further demonstrated that a machine learning approach can be the basis for an efficient panic detection system that can be scaled to a multiuser environment allowing the real-time detection of possible crowd panic events. Future work involves an approach where the real-time data stream will be used to detect such events. Real-time spatiotemporal analysis will investigate the correlation of the geospatial data that have been tagged as stressed and will attempt to highlight simultaneous stress conditions at a crowd level exposing unforeseen event areas that contain subjects with common stress profiles. A smart geo-referenced ERS implementing such functionality can provide the authorities with real-time and accurate information enabling them to respond appropriately.

**Author Contributions:** Conceptualization, Ilias Lazarou; methodology, Ilias Lazarou and Anastasios L. Kesidis; original draft, Ilias Lazarou; review editing, Ilias Lazarou, Anastasios L. Kesidis, George Hloupis, and Andreas Tsatsaris; supervision, Andreas Tsatsaris. All authors have read and agreed to the published version of the manuscript.

**Funding:** This research received no external funding.

**Data Availability Statement:** Publicly available datasets were analyzed in this study. This data can be found here: https://github.com/eliaslazarou/PanicDetectionDataForML, accessed on 4 November 2022.

**Conflicts of Interest:** The authors declare no conflict of interest.

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
