# Peer review of "Panic Detection Using Machine Learning and Real-Time Biometric and Spatiotemporal Data"

_ijgi, doi:10.3390/ijgi11110552_

Round 1

Reviewer 1 Report

This paper focuses on the panic detection online by establishing a sample database, based on heard rate as biometric and with a combination of moving information and gender, age, weight and so on. And then use machine learning to achieve identification. It is interesting. (1)   Authors only make some hypothetical scenarios. Can this information guarantee the panic detection?  Is there a biological basis? (2)  Can the sample of 14 male and 13 female selected by the author guarantee the numerical range of individual feature? How to consider the impact of individual differences on the numerical range?  (3) It needs an explanation why does the window length affect the judgment accuracy because the sliding window is only an observation part of data series.   (4) It is suggested to supplement some detailed processes of machine learning, such as the selection of train samples and test samples, Implementation steps and parameter settings and so on. (5) From the perspective of obtaining implicit information, it is the deep neural network that has a good ability to extract unknown features. It is suggested to supplement the test of this method.

Reviewer 2 Report

The paper suggests a new innovative multimodal dataset for tracking panic events. The dataset consist of time enabled location data and biometrics from wearables and smartphones - analyzed in real-time and used to indicate possible panic attacks for the user. Another purpose of the dataset is for training various ML models so that it becomes easier to distinguish panic attacks from other patterns or symptoms that have other medical reasons.

The main research coming from this paper is related to the field of biometrics with focus on spatiotemporal logging. For some reason the authors insist on discussing Early Warning Systems in the introduction and do this through referring to research within a completely different field that relate to geo-emergencies like earthquakes and environmental hazards.

Then the focus change (last part of introduction and section 2) to be about panic and detection of panic (attacks) Again the paper is not very systematic in the approach towards what their own study is about which results in some detours where we hear about crowd escape patterns and information strategies in case of disasters. It is my observation that the authors struggle to place their own research in the right alley and major parts of the paper tell stories that are not relevant for the field of panic detection in relation to real-time biometric data analysis, which is in this case only about how individuals react to unforeseen events - and reported through biomarkers, sensors in wearables and smart phones. The study is not about crowds and how data collected from large groups can be used to detect unforeseen events!

I think the introduction should be reworked and the same goes for parts of the paper until chapter 3 (methodology). Then the paper change focus towards the proposed datasets and how the study was organized with this this specific purpose in mind. As I read the paper I do not really understand whether raw biometric and positional data was collected and from whome? Everything seems to be more of less generated in the lab with suggested modelling and ML procedures. I think the authors could be more specific in the description of the methodology when they go through the setup of the data collection - who was involved, where did they collect data and what would be sources of errors in the data collection process?

The best part of the paper is definitely chapter 4 with the experimental setup and the profiling of the proposed new features HRMAD. This is at the same time the output from the study and these can still be communicated, but the structure, content and focus of the paper should be revised so that the main results and the purpose of creating a new dataset could be better presented. As it is structured right now it points in too many directions.

L1 - The title actually covers very well the central elements of the paper and should not be changed.

L13 - What are “…through traditional methods.”?

L40 - Why mention systems from a completely different domain - such as earth quakes?

L51 - Please use abbreviations when already introduced terms are presented (early warning systems).

L59 - Must be a title - the first part of this line

L61 - Why mention a specific year when there is no reference?

L75 - Could you please explain what a LSTM approach is? Long Short-Term Memory neural network - preferably with a reference

L89 - You write: “Wearables provide a whole new way of data collection”. In fact wearables have been around for a long time, but technology behind it have evolved dramatically with focus on IoT, sensors, communication of data (4G, 5G…)

L95 - The rapid growing market share of wearables - and other statements that would fit much better in a report about the potential of this technology should not be parts of a scientific paper!

L152 - You have a lot of focus on panic - but in fact other biometrics have focus on other terms such as emotions, and to my knowledge these states do have something in common. It would be good to also mention that and also tell why you do not think that emotions can be used as a marker in the group of psycho-physiological data.

L211 - The system workflow seems more like the workflow of the data into the cloud - but it does not describe the system. You should carefully describe any transformation that data goes through and how communication supports the network. From platform to platform - and how can data again become useful for the user, the scientist or whoever has access to that cloud service

L225 - Maps should show scale, direction, legend and source from where map data come

L231 - Figure 3 You tell here that this is biometric and spatiotemporal data collected from the experiment. Where is the spatial component of this dataset? I can see a timeline but no position, coordinates, distances etc.

L303 - Please try to layout your paper so that tables don’t split on page shift.

L370 - Again a split table between two pages

L477 - References should be checked again and please provide URL’s for the DOI system when possible. Most papers in journals have object identifiers so that it becomes very easy to track the used references. The references are a big mess as they are right now

Reviewer 3 Report

Panic Detection using Machine Learning and Real-Time Biometric and Spatiotemporal Data:

·         Add some of the most important quantitative results to the Abstract.

·         Add/Replace the name of the study area with the Keywords.

·         In the last paragraph of the Introduction, the authors should mention the weak point of former works (identification of the gaps) and describe the novelties of the current investigation to justify that the paper deserves to be published in this journal.

·         Lines 137-142 are not necessary and should be deleted.

·         Discuss more the partially erroneous panic prediction example.

·         At the end of the manuscript, explain the implications and future works considering the outputs of the current study.

·         The quality of the language needs to be improved for grammatical style and word use.

Round 2

Reviewer 2 Report

The authors have followed the advice given in the first round of review and I find the revised manuscript much better structured and easier to read and understand. Thank you for this important contribution. There are only minor corrections left to do here.

Reviewer 3 Report

I appreciate the authors addressing the comments. The manuscript can be accepted in its current form. Congrats!